# Effectiveness and cost-effectiveness of the GoActive intervention to increase physical activity among UK adolescents: A cluster randomised controlled trial

**Kirsten Corder**[1,2], **Stephen J. Sharp**[1,2], **Stephanie T. Jong**[1,2,3], **Campbell Foubister**[1,2], **Helen Elizabeth Brown**[1,2], **Emma K. Wells**[1,2], **Sofie M. Armitage**[1,2], **Caroline H. D. Croxson**[4], **Anna Vignoles**[5], **Paul O. Wilkinson**[6,7], **Edward C. F. Wilson**[8], **Esther M. F. van Sluijs**[1,2]*

1 UKCRC Centre for Diet and Activity Research, University of Cambridge, Cambridge, United Kingdom, 2 MRC Epidemiology Unit, University of Cambridge, Cambridge, United Kingdom, 3 Faculty of Medicine and Health Sciences, University of East Anglia, Norwich, United Kingdom, 4 Nuffield Department of Primary Care Health Sciences, University of Oxford, Oxford, United Kingdom, 5 Faculty of Education, University of Cambridge, Cambridge, United Kingdom, 6 Department of Psychiatry, University of Cambridge, Cambridge, United Kingdom, 7 Cambridgeshire and Peterborough NHS Foundation Trust, Cambridge, United Kingdom, 8 Health Economics Group, Norwich Medical School, University of East Anglia, Norwich, United Kingdom

* esther.vansluijs@mrc-epid.cam.ac.uk

**Data Availability Statement:** Data cannot be shared publicly because of the semi-identifiable nature of the data. Data are available from the MRC

## Abstract

### Background

Less than 20% of adolescents globally meet recommended levels of physical activity, and not meeting these recommended levels is associated with social disadvantage and rising disease risk. The determinants of physical activity in adolescents are multilevel and poorly understood, but the school's social environment likely plays an important role. We conducted a cluster randomised controlled trial to assess the effectiveness of a school-based programme (GoActive) to increase moderate-to-vigorous physical activity (MVPA) among adolescents.

### Methods and findings

Non-fee-paying, co-educational schools including Year 9 students in the UK counties of Cambridgeshire and Essex were eligible for inclusion. Within participating schools ($n = 16$), all Year 9 students were eligible and invited to participate. Participants were 2,862 13- to 14-year-olds (84% of eligible students). After baseline assessment, schools were computer-randomised, stratified by school-level pupil premium funding (below/above county-specific median) and county (control: 8 schools, 1,319 participants, mean [SD] participants per school $n = 165$ [62]; intervention: 8 schools, 1,543 participants, $n = 193$ [43]). Measurement staff were blinded to allocation. The iteratively developed, feasibility-tested 12-week intervention, aligned with self-determination theory, trained older adolescent mentors and in-class peer-leaders to encourage classes to conduct 2 new weekly activities. Students and classes gained points and rewards for engaging in any activity in or out of school. The

Epidemiology Unit(contact via datasharing@mrc-epid.cam.ac.uk) on approval of an analysis plan for researchers who meet the criteria for access to confidential data. The data underlying the results presented in the study are available via datasharing@mrc-epid.cam.ac.uk.

**Funding:** This study is funded by the National Institute for Health Research (NIHR) Public Health Research Programme (https://www.nihr.ac.uk/explore-nihr/funding-programmes/public-health-research.htm; award number: 13/90/18; awarded to: KC, EvS, PW, AV, CC, EW). This work was additionally supported by the Medical Research Council [https://mrc.ukri.org/; Unit Programme number MC_UU_12015/7; awarded to EvS], and undertaken under the auspices of the Centre for Diet and Activity Research (CEDAR), a UKCRC Public Health Research Centre of Excellence. Funding from the British Heart Foundation, Cancer Research UK, Economic and Social Research Council, Medical Research Council, the National Institute for Health Research, and the Wellcome Trust, under the auspices of the UK Clinical Research Collaboration, is gratefully acknowledged [https://www.ukcrc.org/research-coordination/joint-funding-initiatives/public-health-research/; award numbers: 087636/Z/08/Z; ES/G007462/1; MR/K023187/1; awarded to EvS]. This work was also supported by NIHR Biomedical Research Centre Cambridge: Nutrition, Diet, and Lifestyle Research Theme (Grant IS-BRC-1215-20014) to KC, EvS. The funders had no role in study design, data collection and analysis, decision to publish, or preparation of the manuscript.

**Competing interests:** The authors have declared that no competing interests exist.

**Abbreviations:** BMI SDS, BMI standard deviation score; CHU-9D, Child Health Utility 9D; FAS, Family Affluence Scale; MVPA, moderate-to-vigorous physical activity; QALY, quality-adjusted life year.

primary outcome was average daily minutes of accelerometer-assessed MVPA at 10-month follow-up; a mixed-methods process evaluation evaluated implementation. Of 2,862 recruited participants (52.1% male), 2,167 (76%) attended 10-month follow-up measurements; we analysed the primary outcome for 1,874 participants (65.5%). At 10 months, there was a mean (SD) decrease in MVPA of 8.3 (19.3) minutes in the control group and 10.4 (22.7) minutes in the intervention group (baseline-adjusted difference [95% confidence interval] −1.91 minutes [−5.53 to 1.70], $p = 0.316$). The programme cost £13 per student compared with control; it was not cost-effective. Overall, 62.9% of students and 87.3% of mentors reported that GoActive was fun. Teachers and mentors commented that their roles in programme delivery were unclear. Implementation fidelity was low. The main methodological limitation of this study was the relatively affluent and ethnically homogeneous sample.

## Conclusions

In this study, we observed that a rigorously developed school-based intervention was no more effective than standard school practice at preventing declines in adolescent physical activity. Interdisciplinary research is required to understand educational-setting-specific implementation challenges. School leaders and authorities should be realistic about expectations of the effect of school-based physical activity promotion strategies implemented at scale.

## Trial registration

ISRCTN Registry ISRCTN31583496.

## Author summary

### Why was this study done?

- Regular physical activity in adolescence is associated with mental and physical health benefits, but adolescent physical activity levels are low.

- Schools offer a way of promoting physical activity in all adolescents, but interventions need to consider the out-of-school period as well.

- There is limited previous research evaluating adolescent physical activity promotion in large samples with device-measured physical activity and long-term follow-up.

### What did the researchers do and find?

- We conducted a cluster randomised controlled trial of the GoActive intervention, a feasibility-tested physical activity promotion programme co-designed with adolescents.

- After recruiting 2,862 adolescents aged 13–14 years, we found that the GoActive intervention was no more effective than the control condition in preventing declines in adolescent physical activity at 10-month follow-up.

- The process evaluation data show that GoActive was not implemented as intended.

## What do these findings mean?

- Consistent with previous studies, this research-driven approach to school-based physical activity promotion was not effective, with implementation challenges likely playing an important role in the lack of effect.

- Improved understanding of the implementation and delivery challenges of public health interventions in secondary schools is required to improve the effectiveness of physical activity promotion approaches.

## Introduction

Physical inactivity is the fourth largest cause of death worldwide and is thought to be the principal cause of 1 in 3 cases of heart disease [1]. In adolescence, physical activity levels are low. Recent data show that less than 20% of adolescents meet the WHO physical activity guidelines of 60 minutes of moderate-to-vigorous physical activity (MVPA) every day, with little change over time [2]. Not only is inactivity increasingly linked to poor health in childhood [3], it may have long-lasting negative implications for health and educational achievement in adulthood [4,5]. Compared to their inactive peers, active adolescents are more likely to become active and healthy adults [4,6–11], and as such, preventing a decline in activity during adolescence is a major public health priority [1]. The challenge for public health professionals is to identify effective and cost-effective strategies to achieve this.

Evidence suggests that the reduction in physical activity in adolescence predominantly occurs outside of school [12]. School settings offer a way of reaching large numbers of young people from a broad range of backgrounds, and it therefore remains pragmatic and attractive to utilise the school setting for recruitment and delivery of physical activity promotion targeting the whole week [13]. Despite this, physical activity promotion research in adolescent populations is scarce and challenging, with review-level evidence showing no effect on device-measured physical activity and few studies in children over 12 years of age [14,15]. This lack of effect is hypothesised to be due to low intervention fidelity and poor implementation. Studies of the cost-effectiveness of school-based physical activity promotion report mixed results (e.g., [16,17]). As school funders are faced with finite resources, there is a continued need for the identification of effective and affordable school-based activity promotion strategies among older adolescents to inform the best use of limited funds.

Best practice guidelines suggest intervention development should be based on behaviour change theory, existing evidence, and pre-trial qualitative work with the target group [18]. Following our review of existing school-based strategies [19] and novel analyses of existing data [20], we identified limitations of previous adolescent physical activity promotion strategies including a lack of whole-population approaches, limited adolescent involvement in intervention development, poor participant engagement, and lack of consideration of potential negative impacts [13]. We have previously reported on the development and pilot work of the GoActive (Get Others Active) intervention, in which we aimed to address these limitations [13,21]. GoActive employs a population approach, in that it targets a whole year group irrespective of personal characteristics, to overcome the potential stigma of solely targeting at-risk

groups [22], such as adolescents with obesity, or girls. Although GoActive is broadly aligned with self-determination theory [23], our priority was to co-design the intervention with students and teachers. Therefore, we used theory flexibility to enable the incorporation of components strongly suggested in the development work, irrespective of whether they aligned with theory, such as rewards [13].

The objective of this paper is to report on the results of the GoActive cluster randomised controlled trial, which aimed to evaluate the effectiveness and cost-effectiveness of the GoActive intervention to increase whole-day MVPA among adolescents aged 13–14 years.

## Methods

### Study design and participants

The main trial methods have been described in the published protocol paper [24]. All state-run secondary schools in Cambridgeshire and Essex were eligible for inclusion ($n = 103$) and were invited into the study between April and July 2016. The region includes substantial socio-economic diversity and includes both urban and rural areas. In participating schools, school-level written informed consent was obtained from a member of the school's senior leadership team following a meeting between GoActive team members and senior school staff; all students within Year 9 in the 2016–2017 academic year were eligible for inclusion. Ethical approval was obtained from the University of Cambridge Psychology Ethics Committee (PRE.126.2016), and included approval to obtain passive parental consent and written student assent for study participation. The study was prospectively registered (ISRCTN31583496).

Baseline assessments took place early in Year 9 (September 2016–January 2017, with 76% of testing between November and January), the school year in which students become 14 years old. After baseline measurements, participating schools were randomised to the intervention or no-treatment control arm. Allocation used a randomisation list prepared in advance by the trial statistician independent from the measurement team using a random number generator in Stata; 1:1 randomisation was stratified by school-level percentage of students eligible for pupil premium school funding (below or above the county-specific median) and county (Cambridgeshire or Essex). Pupil premium funding, used as a proxy for school-level deprivation, is school funding that aims to reduce effects of deprivation [25].

### GoActive intervention

The GoActive intervention was developed following an evidence-based iterative approach, underpinned by principles central to multiple guidelines and frameworks [26–28], where we incorporated existing evidence and qualitative work with adolescents and teachers [13]. GoActive aimed to increase physical activity through increased peer support, self-efficacy, self-esteem, and friendship quality, and was implemented in tutor groups using a student-led tiered-leadership system. Mentorship and peer-leadership addressed time pressures, which were stated by teachers in our development work as being a barrier to participation in activity promotion programmes, and between-class competition was incorporated as a strategy to encourage teacher enthusiasm [13].

The mapping of intervention components to published behaviour change techniques has been published in previous GoActive papers [24,29], and an overview of key intervention elements and delivery structure is available in S1 Text and S1 Fig. Briefly, each Year 9 tutor group (class or homeroom) chose 2 activities each week from a selection provided. GoActive targeted peer-led class-based activity, with participation also encouraged outside of school. Working with existing class tutors (members of teaching staff), older adolescent mentors encouraged Year 9 students to try at least 1 weekly GoActive session. Activity points were gained for

activity participation in and outside of school irrespective of duration or intensity; students were encouraged to regularly log 'activity points' on the GoActive website to unlock rewards. The GoActive intervention was delivered over 12 weeks. During the first 6 weeks, delivery was facilitated by intervention facilitators (health trainers employed by local councils), who provided school staff and older adolescent mentors with training, support, and resources for intervention delivery. Facilitator support for the programme was reduced during the second 6 weeks to encourage school-led sustainability.

Irrespective of whether students participated in measurements, intervention delivery was at a school tutor group level to all eligible students in intervention schools; parents were encouraged to speak with the school if they wanted to opt their child out of the intervention participation, but no parents chose this option. Control schools received no intervention.

## Outcome assessment

Identical assessment procedures were undertaken at baseline, post-intervention (14–16 weeks post-baseline), and the 10-month post-intervention follow-up in the school. Questionnaire-based measures were also assessed mid-intervention (6 weeks after intervention start). Trained measurement staff conducted measurements using standardised protocols and instruments as detailed in the protocol [24] and summarised in S1 Table. Measurement staff were blinded to allocation, and our dedicated process evaluation researcher independently verified the success of this blinding via email correspondence shortly after the 10-month follow-up measurements.

## Accelerometer-assessed outcomes (including primary outcome)

The pre-specified primary outcome for effectiveness was average daily minutes of MVPA at 10-month follow-up. We measured MVPA at baseline, post-intervention, and 10-month follow-up using wrist-worn activity monitors (Axivity) assessing acceleration (continuous waveform data). Participants were asked to wear the monitors for 7 days continuously, for 24 hours a day, on their non-dominant wrist. These monitors have been validated to assess physical activity energy expenditure [30] and have better wear time adherence and acceptability than commonly used hip-worn monitors among adolescents [31]. Given the 24-hour wear time protocol of the Axivity monitors, a diurnal adjustment was used to reduce any bias caused by imbalances of protocol deviations regarding non-wear [32]. Each day of possible wear was divided into 4 time quadrants: morning (6 AM–12 PM), afternoon (12 PM–6 PM), evening (6 PM–midnight), and night (midnight–6 AM). For participants to be included in analyses, over 6 hours of wear time spread over at least 2 days was required from the possible 42 hours in each day time quadrant (i.e., ≥6 hours from 7 possible mornings, ≥6 hours from 7 possible afternoons, and ≥6 hours from 7 possible evenings). The 'night' quadrant (i.e., midnight–6 AM) was considered as sleep time and was included in the denominator when calculating daily averages of MVPA, for consistency across all participants. Where individuals did not wear the monitor for ≥6 hours at night time, despite the protocol requesting them to wear it continuously for 7 days, average night time values were imputed using population averages ($n$ = 91 at baseline and $n$ = 463 at follow-up), created from GoActive participants with 100% protocol compliance regarding monitor wear, to avoid inflation of MVPA estimates. This method was verified by running simulations excluding night data for a subsample of participants with 100% protocol compliance. For an individual hour to be included for analysis, at least 70% of possible wear time was required, with non-worn time within the hour considered as missing [32].

Monitor output was processed to provide minutes spent in MVPA equivalent to ≥2,000 ActiGraph counts per minute (cpm) [24]. Additional secondary accelerometry-derived

outcomes were average daily minutes of sedentary time (equivalent to ≤100 ActiGraph cpm), average daily minutes of light physical activity (equivalent to 101–1,999 ActiGraph cpm), and average daily activity (represented by average acceleration). In addition to daily averages, all intensity outcomes (including MVPA) were also derived during school time (9 AM–3 PM), during weekday after school time (after 3 PM), and at weekends. Participants who met the inclusion criteria for average daily MVPA were included in any analyses for which they had sufficient data (≥2 days) [17]. As the criteria for deriving average daily MVPA did not require both weekend and weekdays of valid data [33], participant numbers varied by outcome.

## Non-accelerometry secondary outcomes

Student questionnaires were administered at each measurement occasion (baseline, post-intervention, and 10-month follow-up) using measures validated for use in the population. All secondary outcomes were assessed as continuous scores: physical activity self-efficacy (possible score 1–6) [34], social support for activity (1–4) [35], friendship quality (1–5) [36], well-being (1–5) [37], self-esteem (1–4) [38], and self-reported physical activity (0–160) [39]. Anthropometry (height, weight, waist circumference, and bio-impedance to assess body fat percentage) was assessed at baseline and 10-month follow-up by trained staff; BMI standard deviation score (BMI SDS) was calculated from height, weight, age, and gender [40]. BMI SDS was also used to establish weight categories. S1 Table provides further details on assessment and scoring of secondary outcome measures. As a change to the published protocol, anthropometry was not assessed immediately post-intervention to reduce measurement burden on schools and participants and because no meaningful impact on anthropometry was expected short-term.

## Process evaluation measures

The implementation of the programme in each school was assessed through a mixed-methods process evaluation. Full details are available in the published process evaluation protocol [41]. The qualitative component included focus groups with students and mentors; individual interviews with students, facilitators, and contact teachers; and observations of GoActive sessions. Process evaluation questions were embedded into the outcome questionnaires, and were completed by students, mentors, teachers, and facilitators at all follow-up time points. Initial findings from student perspectives were published prior to analysing intervention efficacy to avoid interpretation bias [29], and full triangulation results will be published separately. For the purposes of the current paper, process evaluation questionnaire data were used to assess programme satisfaction (see S2 Table for details). Logging of activity points was tracked using website analytics from the GoActive website.

## Demographic characteristics

Participant descriptive characteristics, including pre-specified effect modifiers (gender, individual socioeconomic position, and ethnicity) were self-reported. Ethnicity was self-reported by participants, who were given 20 response options and additional free text completion options. For descriptive purposes, the reported values were recoded to 5 categories according to recommendations [42] as 'white', 'mixed ethnicity' (identifying with multiple ethnicities), 'Asian' (including South-Asian and Chinese), 'African and/or Caribbean', and 'other'. Ethnicity was subsequently dichotomised for pre-specified moderation analyses ('white' versus remaining categories). Participants completed 6 items from the Family Affluence Scale (FAS) relating to family car ownership, holidays, computers, availability of bathrooms, dishwasher ownership, and having their own bedroom, which were used as a proxy of individual

socioeconomic position by summing answers (possible range 0–13), and dividing into prede-fined affluence groups (low = 0–6, medium = 7–9, high = 10–13) [43,44].

### Economic evaluation

A within-trial cost-effectiveness analysis comparing the GoActive intervention with control was conducted from the perspective of the school funder (i.e., school or local authority bud-get). The reported costs therefore represent the likely costs to a local authority were it to imple-ment the GoActive intervention.

Cost per school and per participant was calculated for intervention group participants and comprised facilitator time input and travel expenses, materials (Quick Cards, sports equip-ment, and rewards and prizes), and teacher time. Staff time inputs were based on the study protocol. Unit costs were based on the mid-point of national pay scales (facilitator and teacher time input), and study financial returns (expenditure on materials and expenses). All costs are reported in 2019 British pounds. There were £0 costs associated with the control arm.

Quality-adjusted life years (QALYs) were assessed using the UK Child Health Utility 9D (CHU-9D), which has been validated for use in adolescents [45] and was included in the par-ticipant questionnaire at baseline, post-intervention, and 10-month follow-up. Total time from baseline to 10-month follow-up, and hence the time horizon for the study, was approxi-mately 2 academic years.

### Sample size

We estimated that 1,310 Year 9 participants would be required to have 85% power to detect a 5-minute difference in change in MVPA between baseline and 10-month follow-up as signifi-cant at the 5% level [24], assuming a standard deviation of MVPA of 17.8 minutes and a corre-lation of 0.59 between baseline and follow-up [21]. Assuming a within-school (intraclass) correlation of 0.034 [46] and 30%–40% loss to follow-up [15,47], we aimed to recruit 16 schools with 150 participants per school.

### Statistical analysis

The statistical analysis plan was approved by the trial steering committee prior to analyses being performed (http://www.mrc-epid.cam.ac.uk/research/studies/goactive/for-researchers/). All analyses were performed using Stata version 15.1 [48]. For MVPA at 10-month follow-up (the primary outcome), the intervention effect, representing the baseline-adjusted difference in change from baseline between the intervention and control groups, was estimated from a linear regression model including randomisation group, baseline value of the outcome (i.e., analysis of covariance [ANCOVA]), and the randomisation stratifiers (pupil premium funding and county). Robust standard errors were calculated to allow for the non-independence of individuals within schools, and the missing indicator method [49] was used to ensure inclu-sion of participants with a missing baseline value of the outcome variable. All secondary out-come variables were analysed using the same method.

For the primary outcome, effect modification by (1) gender, (2) socioeconomic status (medium or low versus high, according to FAS score), (3) ethnicity (white versus any other ethnic background), (4) baseline physical activity (≥60 minutes MVPA/day versus <60 min-utes), and (5) weight status (with underweight or normal weight versus with overweight or obesity) was tested with an F-test of the relevant multiplicative interaction parameter in the ANCOVA model. Effect modifiers were selected based on previous evidence of potential dif-ferential effects [14,15]. Subgroup analyses were performed within all categories defined by these variables.

We conducted a complete-case analysis in which participants and schools were included in the group to which they were randomised, although participants with a missing value of an outcome at follow-up were excluded from the analysis of that particular variable. This is a complete-case analysis that is valid under the assumption that the outcome is missing at random, conditional on randomised group and the baseline value of the outcome [50]. A further analysis of the primary outcome was performed in a per-protocol population, defined as intervention group participants reporting "being active during tutor times at least twice during the last 2 weeks" (i.e., self-reported intervention engagement mid-intervention [week 6 of the intensely facilitated phase of the intervention]) *and* logging activity points on the study website at least once during the whole intervention period. This definition was based on a review of quantitative process evaluation data prior to the main analyses, and reflects the group with highest intervention engagement as opposed to delivery of the protocol with fidelity.

Post hoc sensitivity analyses recommended by the trial steering committee were performed in which the primary outcome was calculated (1) excluding time between midnight and 6 AM and (2) using a stricter inclusion criterion for wear time of 12 hours of wear per quadrant.

Economic analyses comprised calculation of within-trial additional cost per additional daily minute spent in MVPA and additional cost per additional QALY gained over the time horizon. An adjusted analysis included baseline CHU-9D score as a covariate as well as missing data imputed using multiple imputation.

## Results

Fig 1 shows the study flow chart. The team approached 103 schools; most did not respond despite multiple re-contacts. Sixteen schools were initially recruited, 2 dropped out before baseline measurements due to changes in the senior leadership team (1 from Essex and 1 from Cambridgeshire), and replacements were recruited. Of 3,405 Year 9 students eligible for inclusion across all participating schools, 2,862 (84.1%) consented: 1,319 participants at 8 control schools (mean ± SD participants per school: $n = 165 ± 62$), and 1,543 participants at 8 intervention schools ($n = 193 ± 43$). A total of 2,828 (98.8% of those consenting) completed baseline questionnaires, and 2,638 (92.2% of those consenting) had a valid assessment of the primary outcome at baseline. At 10-month follow-up, 2,167 (75.7%) participants attended, and we obtained a valid measure of primary outcome for 1,874 of 2,862 (65.5%) randomised participants. More females and participants with high socioeconomic background, from Cambridgeshire, and with underweight or normal weight provided primary outcome data (S3 Table). Blinding of measurement staff was largely successful (S4 Table); a few cases of unblinding occurred due to student and teacher interaction during measurement sessions.

Baseline characteristics were similar between randomised groups (Table 1). Overall, mean age was 13.2 (SD 0.2) years, 52.1% were male, and 84.7% self-reported as white.

### Primary outcome

Mean accelerometer-assessed MVPA decreased in both randomised groups between baseline and 10-month follow-up. The reduction was slightly larger in the intervention group, although the confidence interval around the intervention effect was wide and inconclusive (Table 2; Fig 2).

### Secondary outcomes

In the whole population, over the duration of the study, overall time spent sedentary increased, and light physical activity decreased (S5 Table). There was no evidence of an intervention effect on average daily accelerometer-based outcome measures post-intervention or at

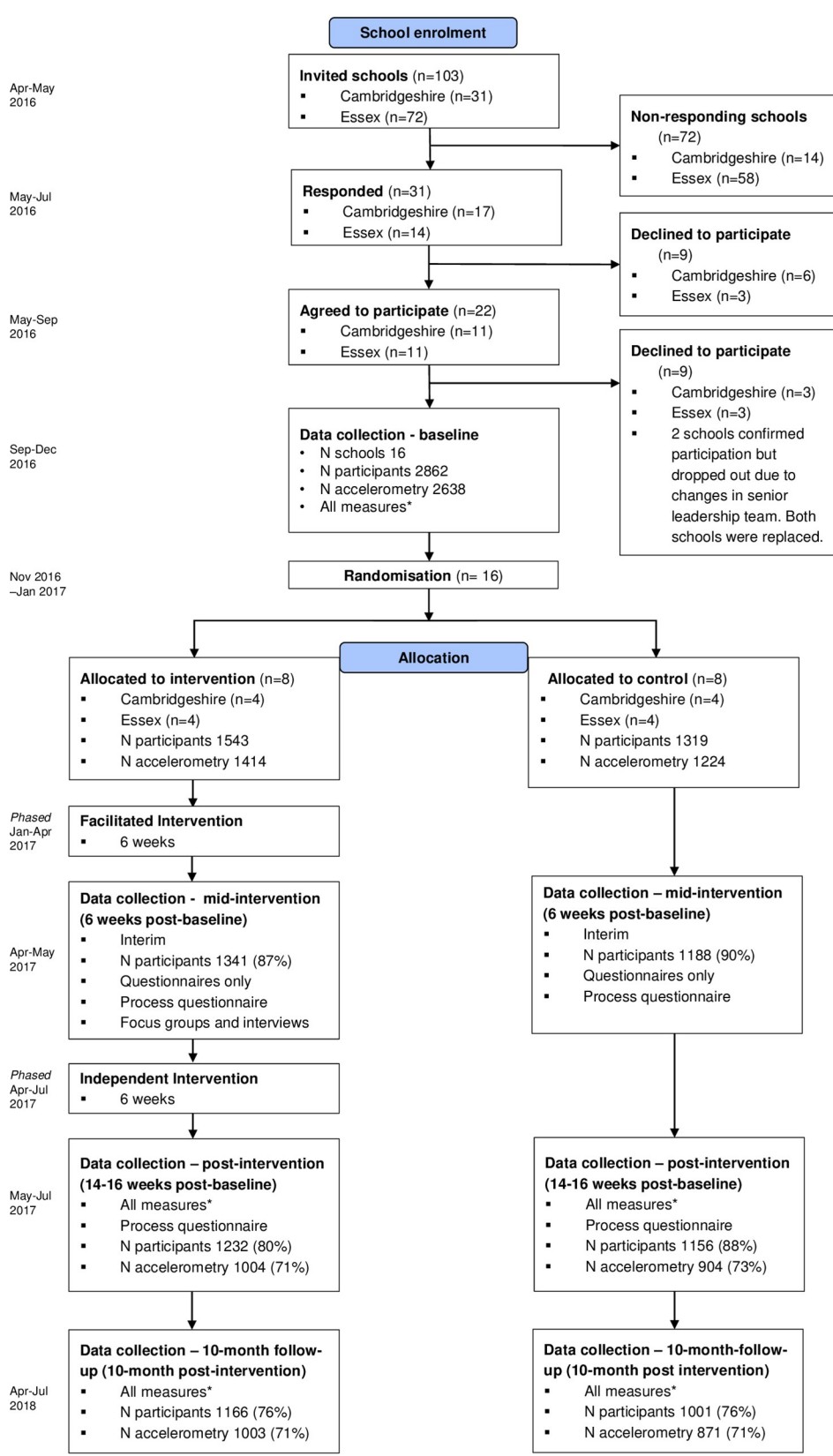

**Fig 1. GoActive study flow chart.**

**Table 1. Baseline characteristics by randomised group: GoActive trial.**

| Characteristic | Control n = 1,319 | | | Intervention n = 1,543 | | |
|---|---|---|---|---|---|---|
| | Percent missing | Mean or percent | SD or n | Percent missing | Mean or percent | SD or n |
| **Age (years)** | 0.0 | 13.2 | 0.4 | 0.0 | 13.2 | 0.4 |
| **BMI SDS** | 0.0 | 0.2 | 1.6 | 0.0 | 0.1 | 1.9 |
| **Body fat (%)** | 3.9 | 20.7 | 10.0 | 5.4 | 20.9 | 9.9 |
| **Waist circumference (cm)** | 0.5 | 70.0 | 9.6 | 0.6 | 70.4 | 9.7 |
| **Gender** | 0.0 | | | 0.0 | | |
| Male | | 53.4% | 704 | | 51.1% | 788 |
| Female | | 46.6% | 615 | | 48.9% | 755 |
| **Ethnicity** | 1.1 | | | 1.3 | | |
| White | | 86.1% | 1,135 | | 83.5% | 1,288 |
| Mixed (identifying with multiple ethnicities) | | 6.2 | 82 | | 6.3% | 97 |
| Asian (including South-Asian and Chinese) | | 3.2% | 42 | | 4.3% | 66 |
| African and/or Caribbean | | 2.2% | 29 | | 2.7% | 41 |
| Other | | 1.3% | 17 | | 2.0% | 31 |
| **Family socioeconomic status** | 0.8 | | | 1.0 | | |
| Low | | 11.0% | 145 | | 16.3% | 252 |
| Medium | | 40.6% | 536 | | 43.4% | 669 |
| High | | 47.6% | 628 | | 39.3% | 606 |
| **Weight status** | 1.4 | | | 2.7 | | |
| With underweight | | 2.6% | 34 | | 2.1% | 33 |
| With normal weight | | 68.5% | 903 | | 66.4% | 1,025 |
| With overweight | | 19.2% | 253 | | 18.5% | 285 |
| With obesity | | 8.3% | 110 | | 10.2% | 158 |
| **County** | 0.0 | | | 0.0 | | |
| Cambridgeshire | | 58.8% | 775 | | 42.4% | 654 |
| Essex | | 41.2% | 544 | | 57.6% | 889 |
| **Pupil premium funding** | 0.0 | | | 0.0 | | |
| Low | | 47.6% | 628 | | 49.2% | 759 |
| High | | 52.4% | 691 | | 50.8% | 784 |

BMI SDS, BMI standard deviation score.

**Table 2. Results for primary outcome of the GoActive trial: Average daily moderate-to-vigorous physical activity (MVPA, in minutes/day) at 10-month follow-up.**

| Measure | Control | | | Intervention | | | Intervention versus control between-group difference, B (95% CI) |
|---|---|---|---|---|---|---|---|
| | Baseline | 10 months | Change from baseline | Baseline | 10 months | Change from baseline | |
| n | 1,224 | 871 | | 1,414 | 1,003 | | |
| Mean (SD) | 35.6 (18.9) | 27.6 (20.6) | −8.3 (19.3) | 35.6 (18.3) | 25.6 (21.5) | −10.4 (22.7) | −1.91 (−5.53, 1.70) |

Between-group difference (intervention effect) is the baseline-adjusted difference in mean change (baseline to 10-month follow-up) in average daily minutes of MVPA between the intervention and control group. Change from baseline calculated based on those with follow-up data (28.8% of control participants and 29.1% of intervention participants had missing data at follow-up). Difference is estimated from a linear regression model including parameters for randomised group (control, intervention), baseline value of the outcome (i.e., analysis of covariance), pupil premium funding (low, high), and county (Cambridgeshire, Essex). Robust standard errors were calculated to allow for non-independence of individuals within schools. Missing indicator method is used to enable participants with a missing baseline value of the outcome to be included in the analysis. Participants with a missing value of the outcome at 10-month follow-up are excluded from this analysis.

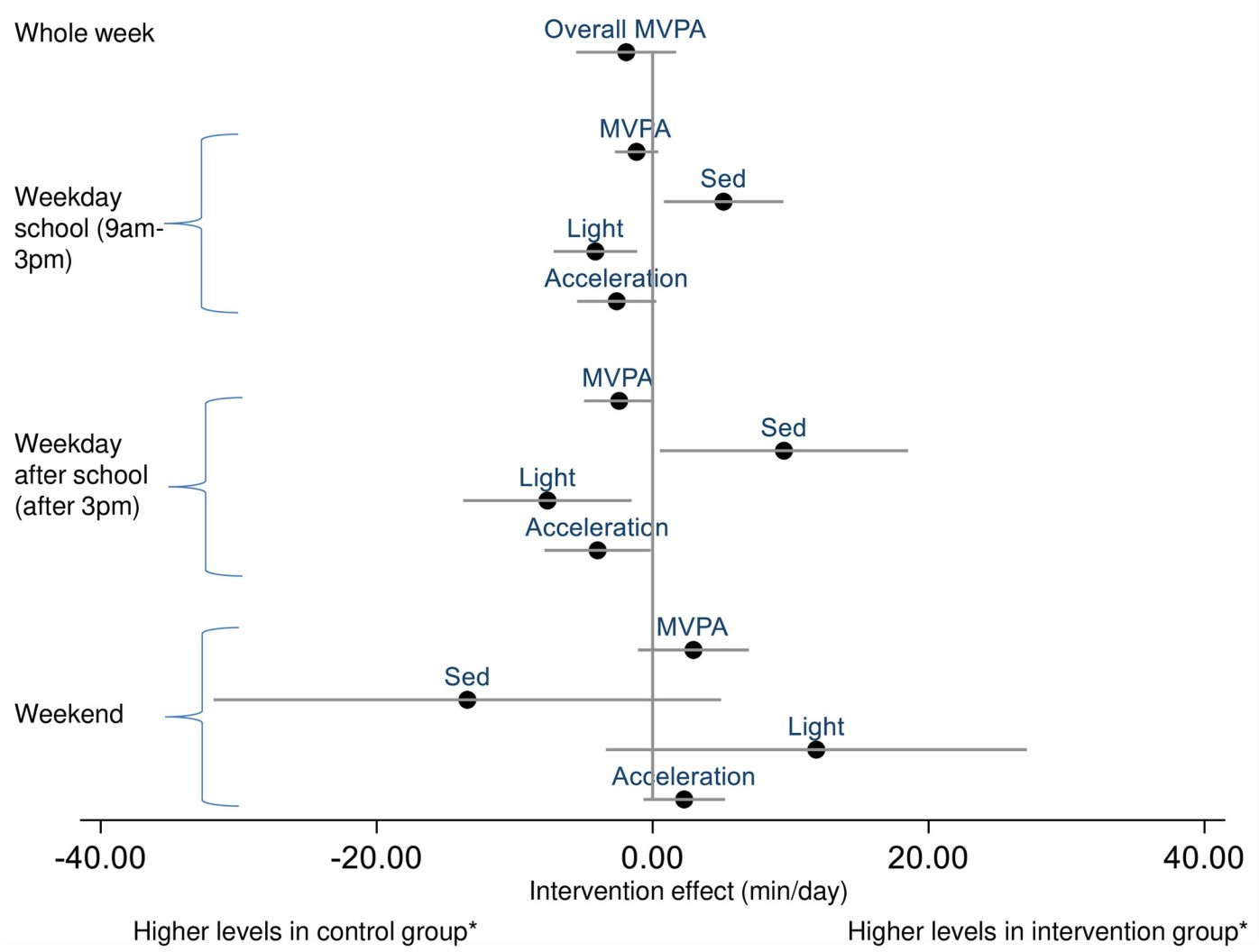

**Fig 2. Intervention effect on continuous secondary physical activity outcomes in minutes per day (acceleration in milli-*g*).** Light, light-intensity activity; MVPA, moderate-to-vigorous physical activity; Sed, sedentary.

10-month follow-up (S6 Table; S7 Table). Time-specific accelerometry-based outcomes showed that on schooldays (weekdays) changes over time were more favourable in the control group (both during school and after school), while at weekends more favourable changes were observed in the intervention group, particularly at 10-month follow-up (Fig 2; see S2 Fig for post-intervention effects and S6 Table and S7 Table for full details).

Self-reported physical activity declined over the duration of the study, whereas little change over time was observed for self-efficacy, social support, friendship quality, well-being, and self-esteem (S5 Table). Overall, the intervention did not affect self-reported outcomes (including assessment of harm assessed using well-being) or anthropometry (Fig 3), with the exception of higher self-efficacy among intervention participants post-intervention (see S8 Table for full analytical results).

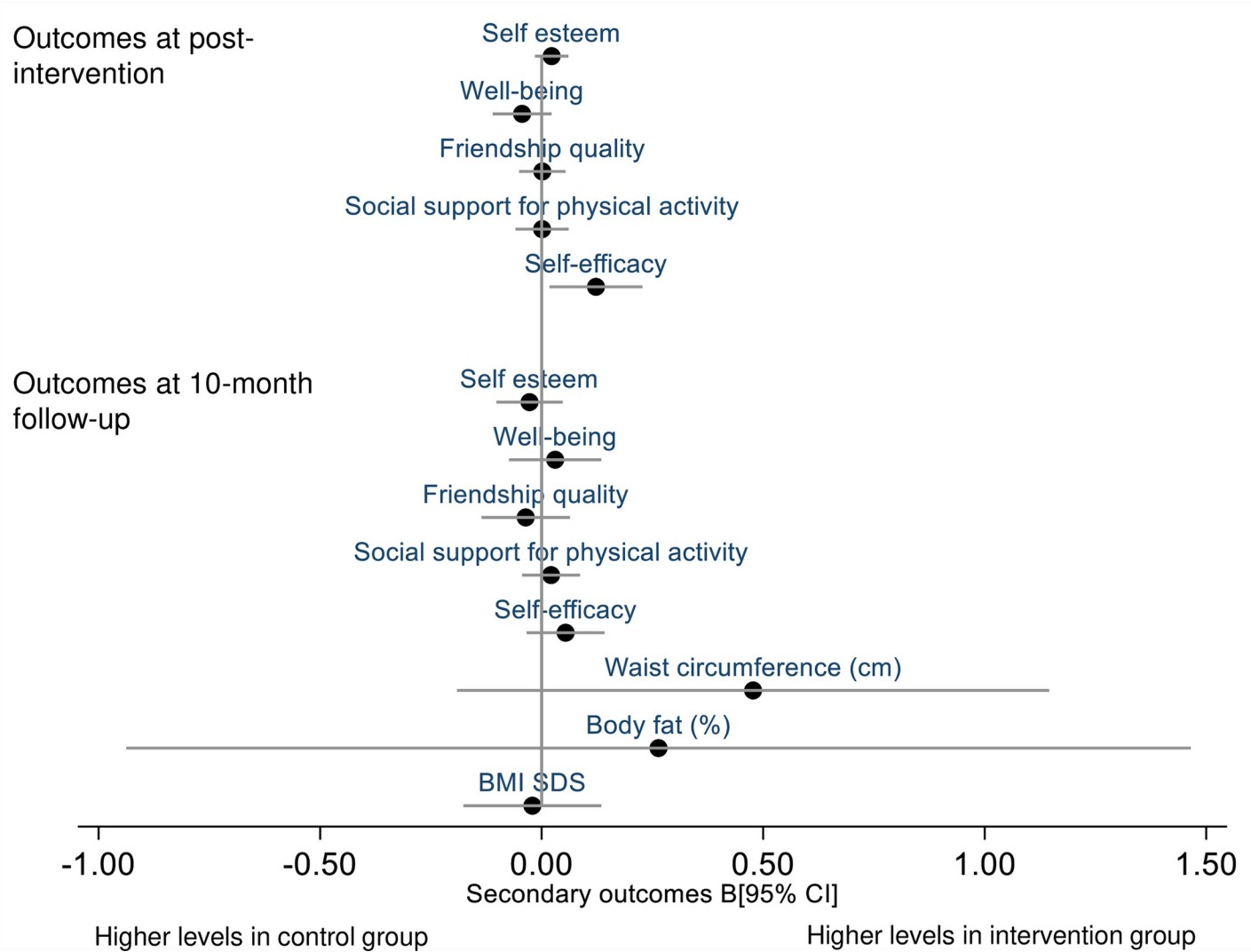

**Fig 3. Intervention effect on secondary psychosocial and anthropometric outcomes presented as baseline-adjusted differences and 95% confidence intervals.** BMI SDS, BMI standard deviation score.

### Effect modification

Tests for effect modification indicated differences in the effect of the intervention between subgroups, in particular between boys and girls, and between high and medium/low socioeconomic status (S9 Table). The results of the subgroup analyses suggested a negative intervention effect among boys and a positive intervention effect for those with low and medium socioeconomic status. However, the subgroup results are inconclusive as confidence intervals included 0 (Fig 4).

### Per-protocol and sensitivity analyses

Only 382 intervention group participants (24.8% of those recruited at baseline and randomised to intervention) met the criteria for inclusion in the per-protocol analysis. The results of the per-protocol analysis did not differ from those of the complete-case analysis (S10 Table). Post hoc sensitivity analyses indicated that results were unaffected by participants with missing data (S2 Text) or different approaches to data processing decisions (S11 Table).

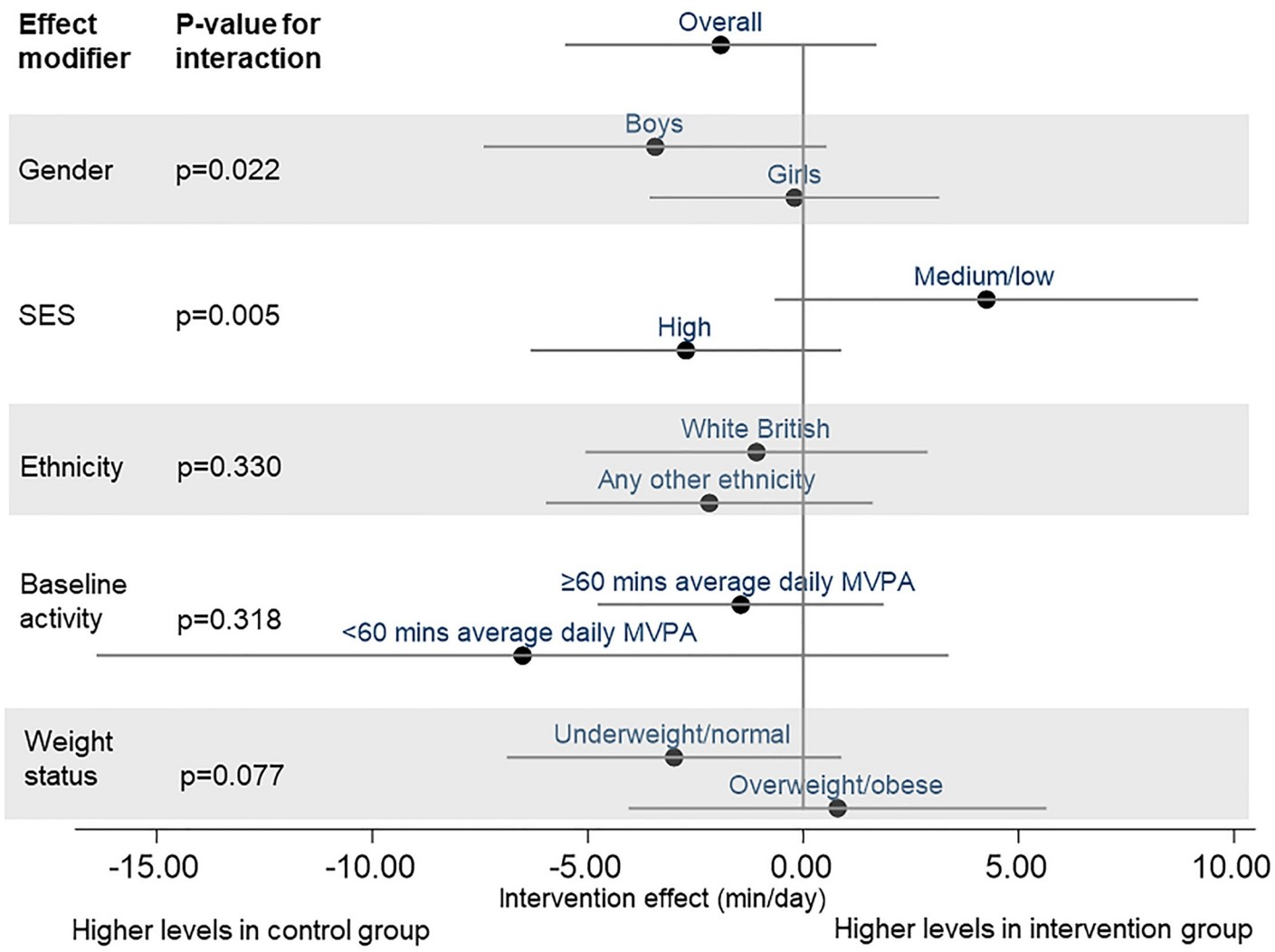

**Fig 4. Intervention effect on primary outcome—overall and within subgroups.** MVPA, moderate-to-vigorous physical activity; SES, socioeconomic status.

## Process evaluation outcomes

Fidelity of the intervention was mixed both within and between schools; 37.9% of students reported attending a GoActive session in the last fortnight post-intervention (ranging from 11.6% to 64.2% between schools). Of students attending baseline assessment and randomised to the intervention group, 46.5% entered activity points using the website. Quantitative data indicated that 7 of 8 intervention schools had mentors, and students at all schools reported having in-class peer-leaders. With regards to satisfaction, 62.9% of students reported that GoActive was fun, 70% of teachers reported that they enjoyed facilitating it, and 87.3% of mentors said it was fun. Session observations and interview data contradicted the effective incorporation of mentors and peer-leaders reported by schools and students. In interviews and focus groups, teachers and mentors discussed that their roles in programme delivery were sometimes unclear. Qualitative data also revealed that the GoActive programme was not consistently implemented within and across schools.

### Adverse events

One participant (in the intervention group) reported an unrelated hospital admission during the baseline measurement period.

### Economic evaluation

The cost of delivering the intervention was estimated to be £2,520 per school compared with control schools; the average cost per student was £13.06 (S12 Table; S13 Table). The mean (SE) QALYs accrued was 1.242 (0.005) in the intervention group versus 1.244 (0.005) in the control group (difference adjusted for baseline data −0.006 [95% CI −0.017 to 0.005]) (S14 Table).

## Discussion

The results of the GoActive trial show that all adolescents became less physically active over time, with no difference between those exposed to the GoActive intervention and those who attended normal school activities. There were inconclusive indications of a more negative effect among boys and a more favourable effect for adolescents with low and medium socio-economic status. Secondary physical activity outcomes showed differential impact across weekdays and weekends, with small between-group differences favouring the control group on weekdays for light physical activity and sedentary time. The findings also indicate that the GoActive intervention is not cost-effective, and that intervention implementation was variable. There was no evidence that the intervention negatively impacted well-being.

Our findings are in line with results from recent reviews suggesting limited effectiveness of research-driven physical activity promotion interventions on whole-day MVPA [14,15]. The absence of intervention effect on time spent in MVPA could be partly due to inadequate implementation; the per-protocol population was small, and our initial process evaluation findings indicate that some intervention components, such as mentorship, were not adequately implemented [29]. However, the per-protocol analysis produced similar results to the main analyses, indicating that if the intervention had been implemented with higher fidelity, it may still not have been effective at a whole-population level. The per-protocol definition focused on website use and reported activity sessions. Use of the website was low and contrasts with the high engagement observed in the pilot trial [21], which indicated preliminary effectiveness. This pattern is common in behavioural interventions, with 75% lower effectiveness seen for behavioural interventions across various health behaviours at the full trial stage compared to feasibility and pilot testing [51]. This is thought to be at least partly due to adaptations needed to implement programmes at scale. Since its inception, GoActive has been designed to be scalable by including a website and flexibility for use in multiple school structures. However, implementation difficulties may have arisen from the provision of implementation flexibility for schools—an issue also identified in the Girls Active study [17]—as well as a lack of clarity in the conceptualisation of the mentor and teacher roles. Additionally, the delivery agent of the intervention changed between the pilot (research staff) and full trial (local-authority-funded health trainer, supported by the research team), which may have contributed to the reduced effectiveness. This points to the challenge for researchers to design interventions that are scalable at the outset, which would minimise the need for major adaptions.

It has been suggested that for a school-based intervention to work, it needs to include a mechanism from at least 1 category outlined in the Theory of Expanded, Extended, and Enhanced Opportunities [52]; the GoActive intervention targeted 2 of these. The 'expansion' mechanism suggests providing new occasions to be active by replacing sedentary time with physical activity, such as adding activity to previously sedentary tutor times. Another suggested mechanism implemented in GoActive is 'extension', which suggests lengthening the time

currently allocated to activity, such as by encouraging students to be active out of school and in tutor times [52]. Process evaluation revealed that the GoActive programme was not consistently implemented and therefore may not have led to sufficient expansion or extension of student activity provision. Low intervention fidelity has implications for the conclusions drawn. If the intervention was either not delivered or not engaged with by students as intended, then no matter how robust the trial design, methods, and analysis, they only give certainty to the findings pertaining to a low-fidelity intervention. As such, in concluding that the intervention was not effective, there is a caveat that it was not effectively delivered.

Secondary outcomes suggested a negative impact of the intervention on light physical activity and sedentary time on weekdays (both in school and out of school), with the opposite seen on weekends. Adolescent-focused process evaluation results indicate that, at times, the intervention may have fostered a climate that was not conducive to physical activity within school (e.g., the sessions appeared to have a lack of social cohesion and connection, and activity choice was often dominated by boys) [29]. However, this may not have extended to weekends. One of the main aims of GoActive was to use school time to encourage participation in activities with friends and family outside of school. On a population level, most of the decline in physical activity during adolescence happens on weekends [12,46]; therefore, it would be worthwhile teasing out what intervention components may be associated with weekend activity. The negative findings for light physical activity and sedentary time on weekdays were reversed for weekends; these opposing associations largely cancelled each other out, leading to no effect for daily averages, with the intervention not appearing to increase activity of higher intensity (i.e., MVPA).

The effect modification analyses suggest that the intervention differentially impacted population subgroups. The intervention appeared to have a more negative effect among boys, as well as those reporting high socioeconomic position. These findings contrast with results from a recent review, which showed no difference between subgroups for intervention effectiveness when assessing whole-day MVPA; however, this was mainly in primary-school-based studies [14]. Across subgroups, our results provide a tentative suggestion of a narrowing of inequalities in physical activity levels, as boys are often reported to have higher activity levels than girls [53], although differences in activity levels by socioeconomic position are less clear [54,55]. The unfavourable impact among boys for average daily MVPA contrasts with our insights from the mixed-methods process evaluation paper exploring satisfaction with the dose received. This evaluation reported higher intervention acceptability among boys, and found that activity choice appeared to be largely driven by boys [29]. These results indicate that gender differences in intervention delivery may not have manifested as expected regarding intervention effect. These contrasting results reinforce the importance of a thorough process evaluation, including observations of delivery, and highlight the complexity of psychosocial issues surrounding activity promotion.

The GoActive intervention appeared to be more effective among individuals with low socioeconomic position, in contrast to a recent meta-analysis showing no differential effectiveness by socioeconomic position [14]. Despite the fact that evidence regarding socioeconomic differences in activity levels is equivocal [55], individuals with lower socioeconomic position may do less vigorous-intensity activity [56] and may have less opportunity for a variety of structured activities [57]. This lack of equity contributes to health inequalities throughout the life course [58], and reducing health inequalities in behaviours and health is therefore a public health priority [59]. It is possible that individuals of low socioeconomic position may have particularly benefited from the chance to try a variety of activities in GoActive as the opportunities may not have been available to them otherwise. There appears to be some utility of comprehensive school physical activity interventions for increasing adolescents' physical activity

behaviour, particularly in disadvantaged neighbourhoods, and such interventions could be particularly relevant among certain population groups [60].

Physical activity across both groups decreased by 10 minutes/day over 2 school years, reflecting the population-level decline seen in physical activity over adolescence [13,61]. Even at baseline, the average activity level of participants was half of the recommended 60 minutes per day, potentially increasing the risk of poor health in the future. It is important to continue to try to increase, or at least prevent the decline of, physical activity among adolescents on a population level, and schools remain a convenient way to reach large numbers of adolescents in one place. However, given limitations in resources and time in school, there may be insurmountable barriers to this approach. UK schools now have very tight budgets, and, given statutory requirements, the additional curriculum time they can allocate to each subject or activity is constrained. Evidence suggests that the majority of the physical activity decline in adolescence occurs out of school, and it has been suggested that the structured nature of the school day may already be somewhat protective of maintaining activity levels [62]. Given the limited success of most school-based interventions in increasing objectively measured whole-day physical activity [14,15], higher level structural changes based on a more in-depth understanding of how physical activity is best integrated in the school, appears increasingly worthwhile.

## Strengths and limitations

We recruited a population representative of the East of England, and our results are relevant to many schools across the UK and to many other high-income settings. Limitations include the adolescent-reported measure of socioeconomic status and the relative lack of participants of low socioeconomic status and non-white participants. However, the percentage of pupils eligible for pupil premium funding in the participating schools was similar to the East of England average (20.9% versus 22.7%) [63]. Moreover, the ethnic diversity of the participants was similar to that of England and Wales (86.1% versus 87.4% white) [64]. Device-measured MVPA as the primary outcome aligns with public health research recommendations for objective and comprehensive evaluation of health promotion programmes [65]. Our recruitment to measurement sessions was high, with 84% of eligible pupils measured at baseline. Although retention on the primary outcome at 10-month follow-up could be perceived as a limitation, we achieved our intended sample size, and the proportion of participants with valid data at follow-up is comparable to that of similar trials [15,47]. To our knowledge, this effectiveness trial was the largest with device-measured physical activity, and addressed many weaknesses of previous trials by including iterative development with the target group and school stakeholders, well-measured pre-specified outcomes, long-term follow-up, detailed process evaluation, and economic evaluation and by having sufficient statistical power to assess effectiveness. However, it is likely that an insufficient dose of the intervention was delivered to achieve the desired effect, and it therefore remains unclear whether the GoActive intervention, if delivered as intended, is effective in changing adolescents' overall MVPA.

## Implications for research

Taken together with recent reviews highlighting the lack of effectiveness of research-driven school-based physical activity promotion strategies [14,15], the current evidence suggests that school-based approaches on the whole do not work to increase adolescent physical activity. However, schools have massive potential to positively impact the health of young people. An overhaul of our approach to secondary-school-based physical activity promotion is needed to encourage school-driven approaches with support from the wider school system, through the use of frameworks such the Comprehensive School Physical Activity Program [66], the World

Health Organization's Health Promoting Schools [67], or the Creating Active Schools Framework [68]. It should be noted, however, that the utility and effectiveness of these frameworks has yet to be established comprehensively. A common feature of these frameworks is the importance of senior leadership buy-in. The GoActive intervention was not initiated by senior leaders, and in most cases their involvement was only for consent sign-off. This may indicate limited buy-in, which may have affected GoActive's potential for effect.

Each school is a unique system with its own culture, and during this research the team experienced barriers to intervention implementation that varied on a school level due to what we often perceived as differences in school culture, ethos, or attitudes [29]. This led us to consider that a randomised controlled trial expecting the same intervention to be replicable, let alone effective, across multiple schools may be an unrealistic expectation and that perhaps aiming for success at a school-by-school level may be more realistic. Although schools are unique microenvironments, standardisation in approaches to every aspect of the curriculum is increasingly becoming normal practice, and appears welcome in schools. There is a need to pursue real and interdisciplinary understanding and collaboration that is likely to deviate from the path of subject-specific research agendas. This should lead to a deeper understanding of the educational system and culture, and may require a shift in the field's ideological principles on physical activity interventions and their delivery in the educational system. Interdisciplinary techniques and disciplines such as ethnography, education, anthropology, sociology, and social networks could progress further understanding of the cultural context of physical activity behaviour in the educational setting.

## Implications for practice

Physical activity promotion initiatives are proliferating throughout schools worldwide without evidence adequately assessing effect or potential harms [69,70]. However, the simplicity of such initiatives has achieved what many designers of complex school-based physical activity interventions aspire to in terms of scale-up, reach, and adoption, and there is also a lot to be learnt from them. Our results from this rigorous and honest evaluation may be uncomfortable, but they highlight the importance of thorough testing of outcomes and unexpected negative consequences and could serve as a warning to those wishing to implement interventions without a candid evaluation. Current research-led approaches to school physical activity promotion do not appear to be effective in their current forms and are unlikely to lead to population-level changes in adolescents' behaviour [14]. The GoActive intervention was rigorously designed with students and teachers and iteratively tested and refined, but despite this rigorous and costly process, when implemented at scale it was no better than the normal school curriculum at preventing declines in adolescent physical activity. We recommend that authorities are cautious about commissioning and rolling out school-based health promotion strategies, that potential unintended negative consequences are considered, and that they are realistic about the scale of behaviour change that can be achieved at a population level and the challenges of implementing a programme as intended.

## Conclusion

The GoActive school-based intervention was not effective in countering the age-related decline in adolescent physical activity. Together with other recent evidence, this suggests that current research-driven approaches to school-based adolescent physical activity promotion are not effective, with implementation challenges likely playing an important role in the lack of effect. Interdisciplinary research should seek to further understanding of the cultural context of physical activity behaviour in the educational setting. Funders, researchers, and local

authorities should be realistic about expectations of the effect of school-based adolescent physical activity promotion strategies implemented at scale.

## Supporting information

**S1 CONSORT Checklist.**
(DOC)

**S1 Fig. GoActive tiered delivery system.**
(DOCX)

**S2 Fig. Intervention effect on continuous secondary physical activity outcomes at post-intervention.**
(DOCX)

**S1 Table. GoActive study outcomes.**
(DOCX)

**S2 Table. Reported items of GoActive process evaluation (post-intervention questionnaires).**
(DOCX)

**S3 Table. Pattern of missing data in the primary outcome (accelerometer-assessed MVPA at 10-month follow-up).**
(DOCX)

**S4 Table. GoActive blinding summary.**
(DOCX)

**S5 Table. GoActive trial primary and secondary outcomes at baseline, post-intervention, and 10-month follow-up.**
(DOCX)

**S6 Table. Secondary outcome results for the GoActive trial: Average daily physical activity (minutes/day) at post-intervention.**
(DOCX)

**S7 Table. Secondary outcome results for the GoActive trial: Average daily physical activity (minutes/day) at 10-month follow-up.**
(DOCX)

**S8 Table. Secondary outcome results for the GoActive trial: Psychosocial and anthropometric outcomes.**
(DOCX)

**S9 Table. Effect modification of the primary outcome, average minutes of MVPA/day.**
(DOCX)

**S10 Table. Primary outcome of the GoActive trial, average minutes of MVPA/day in per-protocol population.**
(DOCX)

**S11 Table. Post hoc sensitivity analyses with different pre-processing decisions regarding primary outcome data.**
(DOCX)

**S12 Table. Protocol-based costing per school per year.**
(DOCX)

**S13 Table. Conversion from cost per school to cost per student.**
(DOCX)

**S14 Table. Quality of life (assessed with CHU-9D): QALYs gained.**
(DOCX)

**S1 Text. Key elements of GoActive intervention.**
(DOCX)

**S2 Text. Impact of deviations from the missing at random assumption on the results for the primary outcome.**
(DOCX)

## Acknowledgments

We thank Active Essex and Everyone Health for providing facilitators for intervention delivery. We are grateful to participating schools and students for their involvement in the study, and we acknowledge GoActive and MRC Epidemiology Unit staff past and present for their involvement in the project.

The views expressed are those of the authors and not necessarily those of the National Institute for Health Research or the Department of Health and Social Care. The funders had no role in study design, data collection and analysis, decision to publish, or preparation of the manuscript.

## Author Contributions

**Conceptualization:** Kirsten Corder, Stephen J. Sharp, Stephanie T. Jong, Helen Elizabeth Brown, Caroline H. D. Croxson, Anna Vignoles, Paul O. Wilkinson, Edward C. F. Wilson, Esther M. F. van Sluijs.

**Data curation:** Kirsten Corder, Stephen J. Sharp, Stephanie T. Jong, Campbell Foubister, Emma K. Wells, Sofie M. Armitage.

**Formal analysis:** Kirsten Corder, Stephen J. Sharp, Stephanie T. Jong, Edward C. F. Wilson.

**Funding acquisition:** Kirsten Corder, Caroline H. D. Croxson, Anna Vignoles, Paul O. Wilkinson, Edward C. F. Wilson, Esther M. F. van Sluijs.

**Methodology:** Kirsten Corder, Helen Elizabeth Brown, Emma K. Wells, Caroline H. D. Croxson, Anna Vignoles, Esther M. F. van Sluijs.

**Project administration:** Campbell Foubister, Helen Elizabeth Brown, Emma K. Wells, Sofie M. Armitage.

**Supervision:** Kirsten Corder, Caroline H. D. Croxson, Anna Vignoles, Esther M. F. van Sluijs.

**Writing – original draft:** Kirsten Corder, Stephen J. Sharp, Edward C. F. Wilson.

**Writing – review & editing:** Kirsten Corder, Stephen J. Sharp, Stephanie T. Jong, Campbell Foubister, Helen Elizabeth Brown, Emma K. Wells, Sofie M. Armitage, Caroline H. D. Croxson, Anna Vignoles, Paul O. Wilkinson, Edward C. F. Wilson, Esther M. F. van Sluijs.

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
