## [Editor Report · Decision Letter 0]

11 Feb 2020

Dear Dr Corder, 

Thank you for submitting your manuscript entitled "Effectiveness and cost-effectiveness of the GoActive intervention to increase physical activity among adolescents: a cluster randomised controlled trial" for consideration by PLOS Medicine.

Your manuscript has now been evaluated by the PLOS Medicine editorial staff [as well as by an academic editor with relevant expertise] and I am writing to let you know that we would like to send your submission out for external peer review.

Kind regards,

Adya Misra, PhD,

Senior Editor

PLOS Medicine

---

## [Decision Letter · Decision Letter 1]

14 Apr 2020

Dear Dr. Corder,

Thank you very much for submitting your manuscript "Effectiveness and cost-effectiveness of the GoActive intervention to increase physical activity among adolescents: a cluster randomised controlled trial" (PMEDICINE-D-20-00312R1) for consideration at PLOS Medicine. 

[LINK]

In light of these reviews, I am afraid that we will not be able to accept the manuscript for publication in the journal in its current form, but we would like to consider a revised version that addresses the reviewers' and editors' comments. Obviously we cannot make any decision about publication until we have seen the revised manuscript and your response, and we plan to seek re-review by one or more of the reviewers. 

We expect to receive your revised manuscript by May 05 2020 11:59PM. Please email us (plosmedicine@plos.org) if you have any questions or concerns.

We look forward to receiving your revised manuscript. 

Sincerely,

Adya Misra, PhD

Senior Editor 

PLOS Medicine

plosmedicine.org

* Please structure your abstract using the PLOS Medicine headings (Background, Methods and Findings, Conclusions).

* Please combine the Methods and Findings sections into one section, “Methods and findings”.

Background- please briefly mention reasons for low physical activity in adolescents

Methods

Please clarify 84% eligible students? 

Please provide additional demographics- like gender?

Please clarify if both Cambridgeshire and Essex schools participated and consider revising the sentence for clarity. 

In the last sentence of the Abstract Methods and Findings section, please describe the main limitation(s) of the study's methodology.

Please replace the interpretation section with conclusions

Abstract Conclusions:

* Please address the study implications without overreaching what can be concluded from the data; the phrase "In this study, we observed ..." may be useful.

* Please interpret the study based on the results presented in the abstract, emphasizing what is new without overstating your conclusions.

* Please avoid vague statements such as "these results have major implications for policy/clinical care". Mention only specific implications substantiated by the results.

* Please avoid assertions of primacy ("We report for the first time....")

Funding information should be moved to the funding section of the article meta-data

The Data Availability Statement (DAS) requires revision. For each data source used in your study: 

Authors do not need to submit their entire data set, or the raw data collected during an investigation. Please submit the following data: The values behind the means, standard deviations and other measures reported; The values used to build graphs; The points extracted from images for analysis.

Introduction

References- please use square brackets throughout

“Compared to their inactive peers, active adolescents are more likely to

become active, healthy and successful adults”- I would recommend removing the word successful here. Your previous work is related to education so please rephrase.

Please include this information in the results section “Although the feasibility and pilot studies were not powered to determine effectiveness, preliminary effectiveness for our

intervention was suggested, with between-arm differences in objectively measured

physical activity at follow-up (difference in means 5.1 min/day (95% CI 1.1 to 9.2)(20).”

This information ought to be in the methods and/or discussion sections “The trial reported here overcame the limitations of much previous research(15) including

iterative development with the target group and school stakeholders, a sample size large

enough to detect a 5-minute difference in the primary outcome between intervention

and control groups, and assessment of long term effects of device-measured physical

activity”

Methods

Could you please mention if the passive mode of parental l consent was approved by the ethics committee?

Role of the funding source- please move this information to the financial disclosure section

Please ensure all questionnaires are cited or provided as SI files 

Discussion

Please present and organize the Discussion as follows: a short, clear summary of the article's findings; what the study adds to existing research and where and why the results may differ from previous research; strengths and limitations of the study; implications and next steps for research, clinical practice, and/or public policy; one-paragraph conclusion.

Please begin this section with “our results show..” or similar, especially at the start of the sentence “The GoActive intervention is not cost-effective”

You mention focus group interviews here but these are not fully articulated in earlier sections. This may not be the focus of the current manuscript but please do mention this briefly in the methods section. 

Reflections and implications sections should be reduced or removed, since much of this is quite speculative. 

Please complete the CONSORT checklist and ensure that all components of CONSORT are present in the manuscript, including how randomization was performed, allocation concealment, blinding of intervention, definition of lost to follow-up, power statement.

Comments from the reviewers:

Reviewer #1: General comments

The manuscript describes findings from the GoActive physical activity intervention for adolescents. Physical activity declines dramatically during adolescence and innovative strategies are needed to engage adolescents in health enhancing physical activity. The GoActive intervention was carefully designed and based on a successful pilot study. Study strengths include the rigorous study design, high quality outcomes, large sample size and high consent rates, inclusion of longer-term follow-up assessments and assessment of cost effectiveness. Despite these strengths, the intervention had no effect on reducing the decline in physical activity typically observed in adolescent populations. The authors provide an honest account of their findings and some useful suggestions for moving the field forwards. I provide a list of minor suggestions for improving the manuscript. 

Specific comments

INTRODUCTION

In general, the authors have provided a strong rationale for their study. Physical activity declines dramatically during adolescence and schools represent an ideal opportunity for intervention. 

METHODS

Study design and participants

The study design and methods are rigorous and appropriate for the study aims. 

Excellent consent rate for students. 

From the 103 schools that were invited, 16 were successfully recruited. Did the research team track reasons for schools' non-participation? Could this information be included in the CONSORT flow diagram (this is mentioned on page 25 of the protocol)?

Were schools match paired according to the four variables before randomisation? See Murray text below: 

Murray, D. M. (1998). Design and analysis of group-randomized trials. New York, NY: Oxford University Press.

Procedures

Self-determination theory (SDT) has emerged as a useful theory for explaining and promoting physical activity in adolescent populations. However, the use of external rewards for participating in physical activity is not consistent with the tenets of SDT for promoting self-determined motivation. Such an approach may have a short-term positive effect, but is unlikely to promote sustained behaviour change. Please comment. 

Statistical analyses

Did the research team conduct two types of analyses (i.e., complete case and intention to treat)? The following sentence appears contradictory- "We conducted a complete-case analysis based on the Intention to Treat principle, in which participants and schools were included in the group to which they randomised, although participants with a missing value of an outcome at follow-up were excluded from the analysis of that particular variable.". From my understanding of ITT, participants should be included in the analysis regardless of whether or not they completed follow-up assessments. Please elaborate. 

Did the authors test for clustering of effects at the class level? Although participants were randomised at the school level, it appears as though aspects of the intervention were delivered during class-time.

RESULTS

If possible, please report the adjusted difference between groups for the psychosocial and anthropometric outcomes in a table (not in a supplementary table). 

DISCUSSION

The discussion provides an honest account of the findings. The authors provide a useful summary of the challenges faced by researchers as they scale-up successful pilot studies. 

The lack of alignment between reward provision and SDT should be addressed in the discussion.

Reviewer #2: See attachment

Michael Dewey

Reviewer #3: This paper reports on the evaluation of impact of the GoActive program on adolescent's total daily MVPA across 10 months, noting that the intervention was the most intensive during the initial 12-week period. The study was conducted in the UK, with a somewhat diverse population but does not necessarily include a large sample of racially/ethnically diverse children, low-income children, nor a high percentage of overweight/obese children, who might generally benefit more from school-based interventions. This was a randomized and blinded study in a large population of children and the authors are to be commended for that. However, throughout the manuscript I found that the writing was not necessarily compelling nor specific. I also think that based on their findings that there is the abrupt and somewhat overstated conclusion that we should give up on school-based PA interventions and that they are not cost-effective. That was not the primary goal of their study and it is indeed unfortunate that the results were not more positive. However, there are questions with regard to some of the analyses, the dose delivered, etc. which might resulted in the inability to find a positive effect on PA. We need to remember that we are working to prevent a natural decline in activity over the childhood years and it indeed may become ever more complex during the adolescent years when academics become more competitive and there more of a loss of light activity and a general increase in sedentary time. Fueling self-efficacy around PA might be indeed very important and maintaining the access to PA. In this study, given the focus on total daily activity it would have also been nice to hear more about the out-of-school time activities of these children - the % engaged in sport, etc. Even data on those who walked to school, vs. other means.

Title: "effectiveness" and "cost-effectiveness" seems a bit redundant/wordy in the title - consider rewording.

Abstract: 

Introduction: "and social disadvantage" seems a bit misplaced - social disadvantage from chronic disease risk or just from alone not being active? 

Methods: 

Should "Year 9" be capitalized? - if so, your second mention is not.

It is not clear what N=165(62) and n=193(43) refers to

Introduction

Overall, the introduction is somewhat disjointed and does not read smoothly. It seems a bit thrown together and does not exactly prepare the reader for where the authors are at in terms of presenting the current work. 

First paragraph

First sentence - do these data represent in a decline in adolescent physical activity from when to present? Or are you simply implying that globally adolescent PA is low - I would clarify be very precise with this compelling data.

Please clarify what "life chances" are

Note that the drop in 5min/day from adolescence to adulthood may not seem that compelling when it is known that children are already dropping on average two minutes per day throughout each progressing year of elementary school….such that levels are already so low. And please just clarify - is this comparing a 17 year old to an 18 year old? Again, some specifics might be more compelling for the reader. 

Paragraph 2

Please cite the reference demonstrating that most PA declines throughout adolescence occur outside of school. This may vary substantially on the socio-demographics.

The thought-stream of this paragraph is not fluid. It is difficult to follow the points that are attempting to be made through discussing all of the reviews - are you attempting to highlight the lack of interventions in adolescence? Or are you high-lighting that school-based interventions are ineffective in increasing total daily activity…or other?

Paragraph 3

Needs a transition to this paragraph shifting the reader to the focus of the current intervention (that is not clearly stated). There are too many ideas/concepts in this paragraph - theory development, prior pilot work, the aim of this study, and why it overcame limitations. Some of that information could even be put in the discussion. Finally, the concept of cost-effectiveness feels a bit thrown in and does not have any background or rationale (although understood) in the text.

Also, please spell out moderate-to-vigorous PA the first time it is used (MVPA) and also be precise that the main outcome is total daily MVPA and not school-time

Methods

Baseline assessments Sept - Jan?

Accelerometry

Why did the criteria for accelerometry data inclusion not include both weekend and weekdays?

Inclusion criteria for 6 hours of wear time over at least two days seems very sparse. Would like to see references to support methodology chosen. The protocol, as written, is difficult to follow.

Reported outcomes

How can individual socio-economic position be self-reported by this group of children?

Anthropometry - understand the subject burden of assessing this measure, however I think it is an unfair statement to say that it was not important to assess at 10 months. There would not be an expectation of weight change, however, the differential impact of programming between weight status categories would be an important outcome to assess and understand.

Economic analysis - more explanation of the measures and costs incurred during the program implementation are needed here.

Process evaluation

Please briefly describe the levels of dose/measures survey/inquired about here and what your main process measures are that are utilized in the present study/analysis.

Analyses

More information is needed on the physical activity survey and how the variables of relative to never active were determined.

Please define what "active during tutor times at least twice during the last two weeks" means. How active? For how long? Is this only during the school day or could it be other? Etc.

Also be more specific about what activity points are and how often children were instructed to log - is this a daily, weekly, or monthly calendar, etc?

Results

In general, the results described are very vague and do not include much data. It is hard to understand the findings/results in the current written form.

Please describe the population in general terms in written results - age, % female, % overweight, % obese, % non-white. 

Primary outcome - results "appearing to favor" seems like very vague language for the results section and should be presented more precisely and with or without significance.

Tests for interaction - similarly the language used is very vague…"suggest"…and results are inconclusive. 

There is general concern that the per-protocol analyses only included 382 children and that the inclusion criteria for following the protocol did not seem to necessitate a very large dose of activity delivered.

Process evaluation

A better understanding of what it means for students to have a peer-leader vs. a tutor, etc. is needed.

Figures. 

Legends need to describe what is being shown in more detail. They currently do not stand alone. What is acceleration?

Is self-efficacy - general or PA self-efficacy?

Discussion

Why does the first paragraph end with the notion that the intervention could have negatively impacted well-being?

If baseline was Sept-Jan then 10month follow-up occurred between the months of? July-November? That doesn't really make sense given the academic year. The discussion also mentions across "two academic years". There was also no mention of accounting for the time of year of measure at baseline vs. followup. (or weather for that matter)

What was the children's average school-time physical activity? How much did this decline over the time-frame? Vs. total daily? And did you indeed see more of a decrease out of school time? I think that the discussion is too focused on weekend days and out of school time, when the intervention was school-based. 

Limitations

Failure to acknowledge some of the variables that could have been assessed such as longer wear-time inclusion criteria for accelerometry, weather, time of year of the different visits, child measure of SES, specified dose of delivery for children to achieve in school, and somewhat of a lack of lower-SES and racial/ethnic diversity. Final there appeared to be a disregard in understanding the dose received and uptake by important sub-groups such as girls and those who are overweight/obese. Discussion of the prevention of decline in minutes of MVPA in girls vs. boys might be very illustrative. 

Given the apparent lack of dose delivered and received by the adolescents in this study, this reviewer believes that it may be somewhat of a premature assumption and editorial by the authors to state that we should divert our resources elsewhere beyond the promotion of physical activity in schools. Multiple modalities are warranted and indeed continue to be very difficult to assess in a natural way, but in a robust clinical trial design with adequate statistical rigor.

Reviewer #4: This paper reports of the effectiveness and cost effectiveness of a large cluster RCT in the UK. Like a number of other well-designed UK school-based trials the intervention aims to use peer leaders to influence younger pupils to become or stay active with the hope of reducing the age-related decline seen. A number of papers related to this study have already been published, including the protocol paper. Despite the preparation for this definitive RCT being linear with clearly related outputs (i.e. development work, feasibility study, protocol paper) the presentation in this paper is not wholly clear. The paper could do with a rework in how the work leading up this this RCT is presented, the clarity with which the actual intervention is presented and the rationale for including process evaluation peppered around the paper. Well done on the whole study and getting it down on paper here, they are a lot of work!

Critical comments:

Process evaluation section is a bit lost in there and doesn't add anything especially if more traiangulation work or reporting is planned in a separate paper. I would suggest removing and just sticking to primary outcomes and the costs.

The authors state the programme was not cost-effective. Firstly, the programme was not effective at changing the primary outcome so cost effective analysis is inappropriate. Should this be a cost consequence analysis? This would describe what the cost consequences of delivering the intervention were? 

Economic analyses of trials like this are in their infancy in the UK. But now we have a number of large NIHR trials including ones from this PHRP call that have used economic analysis (also WAVES) that is building up since methods, tools and data in this area. Have you considered a separate paper on this? One that includes methods, reflections etc.

Major comments:

1. The authors stated the feasibility was promising. What was the difference between the feasibility and this RCT? The authors state that the engagement with online and was low and "contrasts high engagement in the pilot trial" but was there anything related to the delivery that was different? I think this is a key piece of the puzzle of how promising feasibility/pilot does not always translate into effective main trials. Consider (even in a separate short report) a table of differences between feasibility and the main trial in terms of, for example, delivery (type, amount, staff, training), support provided by the research team, the "buy in" from the delivery staff (vs. delivery from vested researchers) etc etc. 

2. Paper goes very quickly from the lit into the background on the trial (top of page 13). A clear opening sentence introducing the programme would be useful there. The section starting with the following blends a little too much with the background/intro to the intervention. "Previous research suggests that activity promotion strategies should be based on behaviour change theory, existing evidence and pre-trial qualitative work with the target group(13, 17). Although the intervention was broadly aligned with self-determination theory(18), our priority was to co-design the intervention with students and teachers. Therefore we used theory flexibility to enable the incorporation of components strongly suggested in the development work, such as rewards." Suggest the whole section be split. Deal feasibility data separately as it is all mixed into one. 

3. In relation to the delivery staff from Active Essex and Everyone Health. How sure are you that the delivery was as robust and fidelity compared to the feasibility? In outsourcing our intervention delivery to suit the funding call/stream (which is a positive to promote sustainability!) are we underserving the original intervention? What did the delivery staff say? What do Active Essex and EH think of the programme, will they keep it on their books?

4. Sentence on "Prior to April 2018, schools received money for every child whose families received income support and had an annual gross income of £16,190 or less(22)." Seems lost and I would say not needed as you have given the reference in the previous statement.

5. Reference number 13 has been used quite a bit and is the reference for the intervention development work. This is not stated anywhere in the paper. I think you need a clearer section about the development of the intervention, the feasibility trial and the feasibility results. This can go in the intro and will help "set the scene" for this definitive cluster RCT. As it stands I am not sure what background info has come from the literature and what has come from the team's own development work. I feel the intro to the development work is undersold!! State that reference 21 is the protocol. 

6. The methods state that in the recruitment areas "substantial socioeconomic diversity and includes both urban and rural areas" existed. Consider whether this is enough to account for the context that schools exist in. "Each school is a unique system" very true. The context is key. Will this be explored in the process evaluation?

7. Results table: I have not seen a %missing column within the table, usually in the footnote. If it has to go in I suggest at the third column. Optional.

8. Costs collection were supplemented by study records, Give examples of what these study records were. Who were responsible for keeping these updated?

9. When describing the primary outcome of PA at 10 months. Be clear that that is the accelerometer measure of PA because the next section talks about self-report.

10. SES of the ppt was used in the sub-group analysis. Please add a justification. Would that mean that if an SES effect was seen that interventions would be tailored depending on a ppt SES?

11. Economic analysis - where do those costs lie with in the current delivery i.e. how much of that cost was bourne by the school (printing, cost of teacher time prepping or delivering), bourne by the research team but presumably would have to be paid for by a commissioner in the future?

12. "There was no evidence that the intervention negatively impacted wellbeing" but this is only based on the measures that were included in Table S1. Restate with caveat. 

13. The paper states that "teachers and mentors found the programme too complicated and we perhaps afforded schools too much flexibility." Does this come out in the PE teacher interviews? This is an interesting point. This tallies with the finding in the Girls Active trial process evaluation whereby "The flexibility created by having choice in activities and timing of delivery on occasion created uncertainty. Without milestones or deadlines teachers found other priorities took over and the programme drifted, potentially explaining why schools did not achieve everything they set themselves in their action plans." Gorely, et al. "Process evaluation of the school-based Girls Active programme." BMC public Health 19.1 (2019). It is a fine balance between them all!

14. The intervention is based on SDT in relation to the adolescent ppts. What about the teachers? Is there an opportunity to explore how this flexibility relates to their need for using their expertise, getting appropriate social support…anything in teacher interviews about this and mapping teacher responses onto a theory. 

15. There is a lot on the PE in the discussion but little in the results. I think this is an error because that process evaluation paper will be standalone and will be good. However, I know it is hard because you want to give readers some context on why the outcomes are the way they are. Perhaps reconsider what and how much of the PE is included herein.

16. I found the results largely hard to follow. The terms "effect modification" and "subgroups" are included the stats section. "Sub-group" and "test for interaction" are in the results. While meditation and interaction analyses are included in the discussion. There needs to be some consistency in how the terms are used and some description up front as to what both are looking to explore in practice. 

17. The authors state "the intervention may have fostered a climate that was not conducive to physical activity within school" (ref 23) Vague, what does this actually mean? 

18. The authors state "our results are relevant to many schools across the UK and to many other high income settings" but what are these results? What is that take home message from this that is relevant?

19. In the reflections section the authors state "the lack of effectiveness of school based physical activity promotion strategies" but these are all the ones that have been in research which are either research-driven. But the PHRP interventions are not always researcher driven. They are often a programme running and then adding research on. There are likely plenty of examples of good practice out there that are school specific and are held by teachers or principals and are part of a whole school-approach to health. Was there any evidence of the buy in from the wider school, other teachers, school senior leadership team?

20. The Daily Mile papers are mentioned and how simplicity of that programme is key. But it is also the cost. The fact that it is free is attractive in times of austerity.

21. The authors state that "Our findings suggest that current research driven approaches to school physical activity promotion are unlikely to be effective" is a sweeping statement. I think taken with the other non-effective trials (even just the others funded through the same NIHR themed call) there is an issue. 

22. Again the statement in the conclusion "Authorities should be cautious about commissioning and proliferating school-based physical activity promotion strategies and be realistic about expectations of effect." Is not representative of this work. 

23. The whole discussion needs tightening up but the comments above may help with that.

24. Is "procedures" the best heading for the section about the intervention? Again, that section needs to be tidied up for clarity. I feel there could be a better use of headings within the methods section to guide readers.

Minor/Typos

1. Is this sentence missing a word "Total accrued time from baseline to 10-month follow-up and hence the time horizon for the study is approximately two academic years."

2. Results. Opening line Figure is misspelled. Called "trial profile"

3. Table 2 title is missing a word

4. S12. What is T4?

5. Figure 4. Medium/low and high. State what subgroup this relates to.

6. Underweight is included in S7 but no mention of that anywhere else. What were the numbers in each category? Add to baseline characteristics.

[LINK]

---

## [Decision Letter · Decision Letter 2]

1 Jun 2020

Dear Dr. Corder,

Thank you very much for re-submitting your manuscript "Effectiveness and cost-effectiveness of the GoActive intervention to increase physical activity among adolescents: a cluster randomised controlled trial" (PMEDICINE-D-20-00312R2) for review by PLOS Medicine.

I have discussed the paper with my colleagues and the academic editor and it was also seen again by xxx reviewers. I am pleased to say that provided the remaining editorial and production issues are dealt with we are planning to accept the paper for publication in the journal.

[LINK]

We look forward to receiving the revised manuscript by Jun 08 2020 11:59PM. 

Sincerely,

Adya Misra, PhD

Senior Editor 

PLOS Medicine

plosmedicine.org

Requests from Editors:

This is a bit unquantified in abstract “Adolescent physical activity levels are low” (also in the AS) please add more specific language 

Please add the trial number at the end of the abstract 

Abstract –please add inclusion criteria briefly

Table 1- please can you change the weight category names to avoid stigmatising labels like overweight, underweight etc and use people first language?

I think there is a simpler way of writing this “The results of the GoActive trial show that adolescents who participated in an evidence based school-level physical activity promotion intervention did not experience less of a decline in accelerometer-assessed physical activity than those who attended normal school activities.” 

We don’t require the funding section within the main text. Please provide this information in the financial disclosure section. The same goes for conflicts of interest

Please remove page numbers from the CONSORT checklist and use paragraphs/sections instead.

Comments from Reviewers:

Reviewer #1: I am satisfied with the authors' revisions. 

Reviewer #2: The authors have addressed all my points. Can we just clear up the point I made about per protocol? On what is now page 20 perhaps after the phrase 'similar results' insert what it is similar to. I thought it meant similar to the last analysis mentioned before which was the per protocol one.

Michael Dewey

Reviewer #4: Thank you to the team for addressing a large amount of review comments so well. I have a few more based on the changes but also a few I may not have caught the first time around.

Introduction

Thanks to the team for reworking the introduction. I find the intro vague. I have a number of comments based on this. 

Line 103. What is "academic achievement in adulthood"?

Line 117. The authors state that "Moreover, evaluation of cost-effectiveness remains rare". Do they mean in school-based trials? In PA research? The reference used to back this up is from 2016. Since then, a number of trials in the UK and wider have included health economics. Suggest this statement is restated or a stronger justification for included health economics is included. E.g. the literature is limited, since a notable paper in 2016 there are more emerging, it is critical to included HE in these studies to understand costs/help decision makers etc etc. 

Line 129. "GoActive employs a whole population approach to overcome the stigma of targeting at risk groups" Whole population? Do you mean the whole of school population? All pupils? Teachers? Population of a town/city? At risk for what? 

Methods

"In participating schools, school-level written informed consent was obtained following a meeting between members of the GoActive team and senior school staff;" Who signed off on the consent? Was it the principal/ head teacher or some other school leader? This is important as I cannot see any other formal involvement of senior school staff in GoActive (but you say that will be looked at in the Process Eval). 

At the first mention of Family Affluence Score please include (FAS) after it as you do use FAS subsequently.

The authors state in a response to Reviewer 3 that "Schools were all randomised at a similar time of year" and that "we do not consider accounting for weather and time of year of visits appropriate in the analyses of a randomised controlled trial." Even though schools might be randomised at a similar time that doesn't mean that their baseline assessments were all scheduled at random times between Sept to Jan. Scheduling was likely based on dates outside of the control of the research team but also not at random. There could be a situation where the majority of intervention schools were baseline tested in Sept/Oct and the majority of control schools baseline testing happened Dec/Jan. Unlikely but possible! Then weather and time of year of visits would be an issue. 

Line 166. Please include a reference for the approach that was followed ("following an evidence-based iterative approach"). Was it the MRC guidelines?

Line 178. "tutor group (class or home room class)" will not make sense to non-UK readers.

I take the authors response comment that they are following CONSORT by abandoning ITT and instead giving a clear description of who was included in this main analysis. The authors also state "This is a complete-case analysis that is valid under the assumption that the outcome is missing at random, conditional on randomised group and the baseline value of the outcome" Can the authors include a reference for this statement of validity of this approach? Is it possible that the missing data are not at random in this type of trial? Have they looked for any differences in key baseline characteristics in those that did provide follow-up data and those that didn't? This also relates to the statement on the loss to follow-up being a limitation. If certain groups (e.g. those OW) provided less data at 10 months then this loss could indeed be a limitation. 

Results 

Line 367. "despite multiple follow-ups" Suggest using a different word for multiple contacts as follow-up appears elsewhere

The sentences in first paragraph of the Results that presents the ppt numbers needs reworking. What is the 193 referring to? What is n-165 referring to? 

I.e. please relook at this section as it need multiple reads to understand that the values in parenthesis are SDs and not references. Perhaps add SD within then parenthesis for each: "2862 (84.1%) consented 371 (8 control, 1319 participants, mean (SD) participants per school: n-165 (62); 8 372 intervention, 1414 participants, n=193 (43)); 2828 (98.8% of those consenting) 373 completed baseline questionnaires, and 2638 (92.2% of those consenting)"

Discussion

Line 580. "However, given resource limitations and time in school limitations, there may be insurmountable barriers to this approach". Can the authors clarify what this means? Where are the resource limitations coming from?

Is it premature to suggest that "through the use of frameworks such as the Creating Active Schools Framework" is the answer to how we overhaul this research area? As the authors point out in the comments section this framework's effectiveness has not been established. I believe it is risky to give one specific example that is new and unproven that could be the panacea to the problem stemming from a number of non-effective trials. Also, this framework is very UK-centric. There are a number of established frameworks out there that have global relevance e,g. WHO https://www.who.int/healthpromotion/publications/health-promotion-school/en/

Is the only level of "school senior buy-in" the fact they sign a consent form. However, in the conclusion the authors state that "School leaders and authorities should be realistic about expectations of effect of school-based physical activity promotion strategies implemented at scale." What does this mean in relation to this study? School leaders were not on board from what I could see so why do the authors think that school leaders have unrealistic expectations (or any expectations) of the effect of these interventions. Same can be said for authorities. Who are these authorities, local authorities? Could the same be said for funders?

Line 654. Is commissioning a PA promotion strategy to schools trying to achieve change at a population level? Is it not a setting level? A population based approach would be something wider and larger like a promo campaign? "realistic about the scale of behaviour change that can be achieved at a population level". The authors may wish to reconsider the use of "population" within this paper. 

In the conclusion the authors state that "that research driven approaches to school-based adolescent physical activity promotion are not effective" yet then go on to call for more research that is interdisciplinary. Consider changing to "the current research driven approaches"

[LINK]

---

## [Editor Report · Decision Letter 3]

1 Jul 2020

Dear Dr Corder, 

On behalf of my colleagues and the academic editor, Dr. Sanjay Basu, I am delighted to inform you that your manuscript entitled "Effectiveness and cost-effectiveness of the GoActive intervention to increase physical activity among UK adolescents: a cluster randomised controlled trial" (PMEDICINE-D-20-00312R3) has been accepted for publication in PLOS Medicine. 

PRODUCTION PROCESS

PRESS

PROFILE INFORMATION

Thank you again for submitting the manuscript to PLOS Medicine. We look forward to publishing it. 

Best wishes, 

Adya Misra, PhD

Senior Editor 

PLOS Medicine

plosmedicine.org